# Elemental Analysis and In Vitro Evaluation of Antibacterial and Antifungal Activities of *Usnea barbata* (L.) Weber ex F.H. Wigg from Călimani Mountains, Romania

**DOI:** 10.3390/plants11010032

**Published:** 2021-12-23

**Authors:** Violeta Popovici, Laura Bucur, Suzana Ioana Calcan, Elena Iulia Cucolea, Teodor Costache, Dan Rambu, Verginica Schröder, Cerasela Elena Gîrd, Daniela Gherghel, Gabriela Vochita, Aureliana Caraiane, Victoria Badea

**Affiliations:** 1Department of Microbiology and Immunology, Faculty of Dental Medicine, Ovidius University of Constanta, 7 Ilarie Voronca Street, 900684 Constanta, Romania; violeta.popovici@365.univ-ovidius.ro (V.P.); victoria.badea@365.univ-ovidius.ro (V.B.); 2Department of Pharmacognosy, Faculty of Pharmacy, Ovidius University of Constanta, 6 Capitan Al. Serbanescu Street, 900001 Constanta, Romania; 3Research Center for Instrumental Analysis SCIENT, 1E Petre Ispirescu Street, 077167 Tancabesti, Romania; suzana.calcan@scient.ro (S.I.C.); iulia.cucolea@scient.ro (E.I.C.); teodor.costache@scient.ro (T.C.); dan.rambu@scient.ro (D.R.); 4Department of Cellular and Molecular Biology, Faculty of Pharmacy, Ovidius University of Constanta, 6 Capitan Al. Serbanescu Street, 900001 Constanta, Romania; verginica.schroder@univ-ovidius.ro; 5Department of Pharmacognosy, Phytochemistry, and Phytotherapy, Faculty of Pharmacy, Carol Davila University of Medicine and Pharmacy, 6 Traian Vuia Street, 020956 Bucharest, Romania; cerasela.gird@umfcd.ro; 6NIRDBS, Institute of Biological Research Iasi, 47 Lascar Catargi Street, 700107 Iasi, Romania; daniela.gherghel@icbiasi.ro (D.G.); gabriela.vochita@icbiasi.ro (G.V.); 7Department of Oral Rehabilitation, Faculty of Dental Medicine, Ovidius University of Constanta, 7 Ilarie Voronca Street, 900684 Constanta, Romania; aureliana.caraiane@365.univ-ovidius.ro

**Keywords:** *Usnea barbata* (L.) Weber ex F.H. Wigg, elemental analysis, metals, antibacterial activity, antifungal activity, disc diffusion method, usnic acid, polyphenols, tannins

## Abstract

This study aims to complete our research on *Usnea barbata* (L.) Weber ex F.H. Wigg (*U. barbata*) from the Călimani Mountains, Romania, with an elemental analysis and to explore its antibacterial and antifungal potential. Thus, we analyzed twenty-three metals (Ca, Fe, Mg, Mn, Zn, Al, Ag, Ba, Co, Cr, Cu, Li, Ni, Tl, V, Mo, Pd, Pt, Sb, As, Pb, Cd, and Hg) in dried *U. barbata* lichen (dUB) by inductively coupled plasma mass spectrometry (ICP-MS). For the second study, we performed dried lichen extraction with five different solvents (ethyl acetate, acetone, ethanol, methanol, and water), obtaining five *U. barbata* dry extracts (UBDE). Then, using an adapted disc diffusion method (DDM), we examined their antimicrobial activity against seven bacterial species—four Gram-positive (*Staphylococcus aureus*, *Enterococcus casseliflavus*, *Streptococcus pyogenes*, and *Streptococcus pneumoniae*) and three Gram-negative (*Escherichia coli*, *Klebsiella pneumoniae*, and *Pseudomonas aeruginosa*)—and two fungi species (*Candida albicans* and *Candida parapsilosis*). Usnic acid (UA) was used as a positive control. The ICP-MS data showed a considerable Ca content (979.766 µg/g), followed by, in decreasing order, Mg, Mn, Al, Fe, and Zn. Other elements had low levels: Ba, Cu, Pb, and Cr (3.782–1.002 µg/g); insignificant amounts (<1 µg/g) of Hg and V were also found in dUB. The trace elements Ag, As, Cd, Co, Li, Tl, Mo, Pd, Pt, and Sb were below detection limits (<0.1 µg/g). The DDM results—expressed as the size (mm) of the inhibition zone diameter (IZs)—proved that the water extract did not have any inhibitory activity on any pathogens (IZs = 0 mm). Gram-positive bacteria displayed the most significant susceptibility to all other UBDE, with *E**nterococcus casseliflavus* showing the highest level (IZs = 20–22 mm). The most susceptible Gram-negative bacterium was *Pseudomonas aeruginosa* (IZs = 16–20 mm); the others were insensitive to all *U. barbata* dry extracts (IZs = 0 mm). The inhibitory activity of UBDE and UA on *Candida albicans* was slightly higher than on *Candida parapsilosis*.

## 1. Introduction

Plants have been used in ethnomedicine since ancient times due to their numerous pharmacological activities. Over the years, the benefits of various plant extracts and natural compounds to maintain good health status, prevent disease, and ameliorate different pathologies have been confirmed [1]. The Traditional Medicine Division of the World Health Organization [2] highlights that the thousand-year-old use of medicinal plants as therapeutic resources should be considered due to their efficacy [3].

In the large world of plants, lichens are symbiotic organisms involving a fungus (mycobiont) and autotrophic partner—alga or cyanobacteria (photobiont) [4]. Photobionts are surrounded by mycobiont hyphae, which constitute around 90% of the total thallus biomass; structurally and functionally, this association is very different from free-living algae/cyanobacteria and fungi [5]. In lichen symbiosis, the mycobiont provides a suitable environment for the photobiont’s growth and for protection against intense irradiation, microorganisms, and herbivores. Besides a wide range of specific organic compounds (known as lichen secondary metabolites [6] with various bioactivities [7]), a mycobiont supports photobiont metabolism, assuring its required minerals [4]. Lichens’ mineral nutrition mainly depends on atmospheric sources and has limited water and gas exchange [8]. These characteristics make them significant air pollution biomonitors [9,10] or environmental risk detectors [11,12]. The lichen thallus effectively absorbs minerals from wet and dry atmospheres [13]; it also contains metal particles from various substrates [14,15]. The minerals’ distribution within lichen thalli is not homogenous and depends on their morphological properties [4]. Thus, photobionts retain only the metals used for their metabolic processes (Ca, Mg, Co, Cu, Fe, Mn, Mo, Ni, V, and Zn) [16]. These metals have various roles in maintaining lichen homeostasis.

In addition to the metals required for photobiont survival and assuring metabolic processes, lichens can accumulate and retain many heavy and trace metals in quantities that vastly exceed their physiological requirements. They tolerate these high metal concentrations by sequestrating metals in the extracellular space as oxalate crystals, after the mycobiont’s production of organic short-chain acids (oxalic, citric, and malic) [17]. Complexing them with the lichen’s secondary metabolites, such as phenolic acids [18] (depsides, depsidones [19], and dibenzofurans [20]), leads to another storage form of metals in this space.

Heavy and trace metals in the lichens’ photobionts generate metal stress by diminishing chlorophyll, photosynthesis rate, photosystem II (PS II) quantum yield, and inducing changes at the cellular and tissue levels [21]. They are often involved in oxidative stress by generating reactive oxygen species (ROS) [22]. Therefore, excess metals must be excluded from the cytosol and removed through efflux or compartmentalization to preserve normal metabolic functioning. Antioxidants, phytochelatins transporting metals to the extracellular space, and metallothioneins with metal-binding properties [21] represent lichens’ most known defense mechanisms.

Lichens’ secondary metabolites also have numerous bioactivities: antioxidant, anti-inflammatory, anticancer, cytotoxic, antigenotoxic, antimutagenic, antibacterial, antifungal, and antiviral [23,24]. Nowadays, these specific natural compounds display significant roles in modern drug development, especially for antimicrobial agents, due to a significant emergence of multidrug-resistant (MDR) pathogenic microorganisms. The evolution of multidrug antimicrobial resistance in commensal bacteria [25] has become a prominent public health concern. Therefore, the progressive loss of efficacy of conventional anti-infective treatments represents a high challenge for herbal medicine to develop drugs with a broad spectrum of antimicrobial activity and lower side effects [26].

Deciphering the antibiotic resistance mechanisms developed in pathogenic microorganisms, Blanco et al. (2016) described two principal ways of acquiring resistance: reduction of the microbial affinity for the antibiotic (mutations in genes encoding the antimicrobial targets) or diminution of the active concentration of the antibiotic inside the cell [27]. For the second pathway, they tried to classify the biochemical mechanisms of MDR into three categories: production of hydrolytic or modifying enzymes; mutations in the antibiotics’ transporters, impeding their cell penetration; and energy-dependent efflux pumps to extrude the antibiotics, blocking their access to the target and also generating cross-resistance to numerous antimicrobial drugs [27]. Furthermore, they proved that various MDR efflux pumps exist in Gram-positive bacteria (*Staphylococcus aureus* and *Streptococcus pneumoniae*), Gram-negative bacteria (*Pseudomonas aeruginosa* and *Escherichia coli*), and fungi (*Candida albicans)* [27]. Furthermore, *Streptococcus pyogenes* [28], *Enterococcus casseliflavus* [29,30], *Klebsiella pneumoniae* [31,32], and *Candida parapsilosis* [33] were also included in MDR pathogens, according to other studies.

Many researchers proved that the extracts of *Usnea* lichens in different solvents had inhibitory activity on pathogens known for antimicrobial resistance [34,35]. As a representative of the *Usnea* genus, *U. barbata* has been used for thousands of years to treat various diseases, including infectious ones [36]. Its phytochemical profile [37] displays bioactive secondary metabolites as specific phenolic compounds (depsides and depsidones), dibenzofurans (usnic acid) [38,39], and diphenyl ethers [40,41]; various representatives of these structural categories proved to have significant antibacterial and antifungal properties.

Recently, a research team examined *U. barbata* as a biomonitor of element deposition in the southern Patagonian forest connected with the Puyehue volcanic event [42]. Our study proposes an ICP-MS analysis for 23 metals on *U. barbata* native lichen from an unpolluted forest zone of volcanic mountains (Călimani Mountains, Suceava County, Romania). This lichen species was harvested from a coniferous forest in a peat bog zone with acid soil; it is important to mention that the *U. barbata* habitat zone is one of Romania’s richest natural peat areas [43].

Using an agar diffusion method, another previous study using acetone extracts of six *Usnea* sp. investigated their antimicrobial effects against seven bacterial species [44]. Using the green chemistry concept, we obtained five UBDE in “preferable” solvents—acetone (UBA), ethyl acetate (UBEA), methanol (UBM), ethanol (UBE), and water (UBW) [45,46]—and evaluated their antibacterial and antifungal properties on seven bacterial and two fungal species by an adapted disc diffusion method. Then, the obtained results were analyzed in correlation with the phytoconstituents of each *U. barbata* dry extract.

## 2. Results

### 2.1. Elemental Analysis

Twenty-three metals were analyzed in dried *U. barbata* lichen, and only thirteen elements were detected; the content of ten of the metals was below the quantification limit (LOQ) value. The LOQ values were in the range of 0.1–5.0 µg/g. Five elements had LOQ = 5 µg/g (Ca, Fe, Mg, Mn, and Zn), one element (Al) had LOQ = 1.0 µg/g, and seventeen others had LOQ = 0.1 µg/g. The obtained results are summarized in Table 1. Other detailed data were added in the Appendix A.

The dried lichen showed a significant calcium (Ca) content (979.766 ± 12.285 µg/g), followed in decreasing order by magnesium (Mg, 172.721 ± 0.647 µg/g), manganese (Mn, 101.425 ± 1.423 µg/g), aluminum (Al, 87.879 ± 1.152 µg/g), iron (Fe, 52.561 ± 2.582 µg/g), and zinc (Zn, 20,536 ± 0.125 µg/g).

Other elements, such as barium (Ba, 3.782 ± 0.052 µg/g), copper (Cu, 1.523 ± 0.013 µg/g), lead (Pb, 1.296 ± 0.007 µg/g), and chromium (Cr, 1.002 ± 0.008 µg/g) had low levels; mercury (Hg, 0.671 ± 0.020 µg/g), nickel (Ni, 0.449 ± 0.011 µg/g), and vanadium (V, 0.241 ± 0.004 µg/g) were present in insignificant amounts in dUB (Table 1).

Finally, data summarized in Table 1 show that ten elements—silver (Ag), arsenic (As), cadmium (Cd), cobalt (Co), lithium (Li), thallium (Tl), molybdenum (Mo), palladium (Pd), platinum (Pt), and antimony (Sb)—were non-detected; their content (µg/g) was lower than the quantification limit (LOQ) value (<0.100 µg/g).

The spike recovery (%) values for all elements were included in the admissible range of 70—150%, proving the specificity of our ICP-MS analysis (Appendix A).

### 2.2. Characterization of Usnea barbata (*L.*) Weber ex F.H. Wigg Dry Extracts

The results obtained in our previous studies [46] performed on all five *U. barbata* dry extracts allow the characterization and comparative analysis of these extracts, both in terms of extraction conditions and the content of secondary metabolites (Table 2) correlated with the used solvents.

Data from Table 2 show that dried lichen extraction in both alcohols (ethanol and methanol) had the highest yields—12.52% and 11.29%, respectively, followed by, in decreasing order, acetone (6.36%) and ethyl acetate (6.27%); the water extract was obtained with the lowest yield (1.98%). Analyzing the content of secondary metabolites in all *U. barbata* dry extracts, it can be observed that UBEA had the highest amounts of usnic acid (376.73 mg/g) and tannins (24.40 mg PyE/g), and the lowest TPC (42.40 mg PyE/g). The dry acetone extract shows the highest polyphenol content (101.09 mg PyE/g) and a significant UAC (282.78 mg/g); tannins are found in UBA in a low concentration (3.85 mg PyE/g). The *U. barbata* dry extracts in ethanol and methanol displayed the secondary metabolites in similar amounts. The UBE has 127.21 mg/g UA and 67.3 mg PyE/g polyphenols; in UBM, UAC = 137.60 mg/g and TPC = 70.70 mg PyE/g. Tannins were also quantified in both alcohol extracts as follows: 14.70 mg PyE/g in UBE and 9.99 mg PyE/g in UBM. Finally, the data summarized in Table 2 show that usnic acid was non-detected in the *U. barbata* water extract; UBW also has 45.80 mg PyE/g polyphenols and the lowest TC (1.31 mg PyE/g).

### 2.3. Antibacterial Activity

As a significant secondary metabolite in *Usnea barbata*, responsible for various bioactivities, usnic acid was used as a positive control.

For each filter paper disc impregnated with 10 µL of the sample solutions, the diffusible amounts of UA and various UBDE were calculated, and 1290–1620 µg is the range of these values, mentioned in Table 3. For selected antibiotics as positive controls, diffusible amounts were 5.0 µg Levofloxacin (LEV) and 30 µg Tetracycline (TET). The solvent for UA and UBDE (0.1% DMSO), used as a negative control, showed no inhibitory effect on the bacterial strains’ growth. Therefore, IZs values (mm) from Table 3 indicate the bioactivity of UBDE and UA exclusively. After 24 h incubation, UBW did not have any inhibitory action on Gram-positive nor Gram-negative bacterial growth (IZs = 0 mm).

On Gram-positive bacteria, UA and all other UBDE (in ethyl acetate, acetone, ethanol, and methanol) variously inhibited the growth of bacterial colonies (Table 3).

Thus, on *Staphylococcus aureus* (*S. aureus)*, IZs (mm) increased in order: 11.66 ± 0.94 mm (UBE), 12.66 ± 1.24 mm (UBA), 13.00 ± 0.82 mm (UBM), 14.33 ± 0.94 mm (UBEA), and 16.00 ± 0.82 mm (UA). Our results showed that *S. aureus* had intermediate susceptibility (directly proportional with the dose) to UA, compared to both antibiotics (IZs >16 mm), and resistance to all UBDE (IZs ≤ 14 mm).

*Enterococcus casseliflavus* (*E. casseliflavus)* manifested the most significant susceptibility to UA and UBDE, compared with both antibiotics (IZs > 19 mm, Table 3). The methanol extract had the highest antibacterial effect (IZs = 22.00 ± 0.82 mm), followed by all the other UBDE with similar IZs values (around 20 mm).

On *Streptococcus pyogenes (S. pyogenes)*, interpretation of the obtained data was performed in comparison with Levofloxacin. Our results show that *S. pyogenes* was susceptible only to UBM (IZs = 20.00 ± 1.63 > 17 mm), and resistant to UA and other UBDE (IZs ≤ 13 mm).

Compared to the same antibiotic, *Streptococcus pneumoniae* (*S. pneumoniae*) proved to be susceptible to UA, UBEA, UBE, and UBA (IZs ≥ 17 mm) and resistant to UBM (IZs ≤ 13 mm). Both alcohol extracts (UBE and UBA) showed similar activities (IZs values were 18.00 ± 1.63 mm, and 18.00 ± 0.82 mm, respectively), being higher than UA (17.00 ± 1.63 mm) and UBEA (17.00 ± 0.82 mm).

The Gram-negative bacteria displayed very different levels of susceptibility after 24 h of incubation with UBDE (Table 3).

Therefore, *Pseudomonas aeruginosa* (*P. aeruginosa*) highlighted the most considerable level of susceptibility to UBE (IZs = 20.00 ± 1.63 mm) and UBM (IZs = 19.67 ± 1.25 mm), compared to both antibiotics (IZs ≥ 19 mm). Compared only to Levofloxacin, *P. aeruginosa* was also susceptible to UBA (17.00 ± 0.82 mm) and UBEA (17.33 ± 2.05 mm)—with IZs ≥ 17 mm and intermediate susceptibility, dose-dependent, to UA (IZs = 16.00 ± 0.82 mm).

*Escherichia coli (E. coli*) was highly resistant to UA (IZs = 7.00 ± 0.82 < 11 mm) and all UBDE (IZs = 0 mm). Finally, UBDE and UA did not show any inhibitory effects (IZs = 0 mm) on *Klebsiella pneumoniae (K. pneumoniae*).

### 2.4. Antifungal Activity

Antifungal effects of UBDE were evaluated on two *Candida* species, *Candida albicans* (*C. albicans*) and *Candida parapsilosis* (*C. parapsilosis*) (Table 4). Two known antifungal drugs, Fluconazole (FLUCZ) 25 µg and Voriconazole (VORI) 1 µg, were used as positive controls.

*C. albicans* displayed an intermediate susceptibility, dose-dependent, after 24 h of incubation with UBM (16.33 ± 2.05 mm) and UBE (15.33 ± 1.24 mm), compared to both antifungal agents (Table 4). UBA (13.00 ± 1.63 mm) and UA (10.00 ± 0.82 mm) also had inhibitory effects on *C. albicans*, but compared with Voriconazole (I = 16–14), both IZs ≤ 13 mm were included in the resistance zone. Finally, *C. albicans* was also highly resistant to UBEA and UBW (IZs = 0 mm).

Otherwise, *C. parapsilosis* showed significant susceptibility only to UA action (20.00 ± 1.63 mm) compared to both antifungal drugs (IZs ≥ 19 mm). It also had considerable resistance to all UBDE (Table 4). Hence, only UBEA showed low IZs (7.00 ± 0.82 < 13 mm), and all other UBDE had no inhibitory effects (IZs = 0 mm).

The influence of the metabolite content on antimicrobial activity was evaluated by performing linear trendlines with linear equations and comparing the correlation coefficient values (R^2^). The significant results (R^2^ > 0.5) are summarized in Table 5.

Thus, *U. barbata* dry extracts’ antibacterial effects—expressed as IZs (mm)—against *S. aureus* and *S. pneumoniae* are moderately correlated with usnic acid content (R^2^ = 0.6187). UBDE inhibitory activity against *S. pneumoniae* also proved to have a moderate correlation with UAC (R^2^ = 0.5571).

The data from Table 5 also show the correlation between antifungal activity of UBDE and secondary metabolite content. On *C. albicans*, the inhibitory effect moderately correlated with TPC (R^2^ = 0.5523). On *C. parapsilosis,* the UBDE inhibitory effects were moderately correlated with the other two metabolites, UAC (R^2^ = 0.5342) and TC (R^2^ = 0.6766).

## 3. Discussion

Macroelements, such as Ca and Mg, are highly represented in lichens [9,47], due to their role in photobiont metabolism. Other trace and heavy metals are contained in lichens in various amounts, depending on atmospheric and soil conditions [11,48,49]. Therefore, the elemental composition of *U. barbata* from the Călimani Mountains results from the habitat zone-specific properties. The Călimani Mountains are the highest Romanian volcanic mountains, and the coniferous forest soil is adjacent to the Tinovul Mare Poiana Stampei peat bog [43]. This peat bog has a natural origin, its accumulation beginning in the post-glacial period; the soil color is tawny (brown) due to humic compounds and peat particles in suspension [43]. The particular conditions of the *U. barbata* native zone consist of seasonal water level fluctuations with thermic variations between −1 °C and 14 °C, pH values being 3.6–5.0 [43]. The precipitation range in this zone is 600–800 mL [43]. Cazacu et al. (2018) explored this zone, extracting soil samples and performing pH measurement and trace metal analysis through X-ray fluorescence spectrometry [43]. The data obtained showed pH values between 4.09 and 5.89 and several trace/heavy metals: 45.93 µg/g Cr, 18.64 µg/g Co, 22.14 µg/g Ni, 23.56 µg/g Cu, 87.61 µg/g Zn, 0.31 µg/g Cd, 41.3 µg/g Pb, and 10.99 µg/g As [43]. Their amounts were higher than those mentioned in the national protocols, but no values exceeded the alert threshold [43]. Our elemental contents in dried *U. barbata* lichen were as follows: 1.002 µg/g Cr, 0.449 µg/g Ni, 1.523 µg/g Cu, 20.536 µg/g Zn, and 1.296 µg/g Pb; Co, Cd, and As were undetected because their contents were <0.100 µg/g. Both groups of trace/heavy metals values were correlated and presented in Figure 1.

*Usnea barbata* and other lichen species can be transplanted from native zones to polluted zones for biomonitoring reasons [50,51]. Several studies mentioned different *Usnea* sp. used in this scope and compared with other lichen species. Thus, Bergamaschi et al. (2007) proved that *U. hirta* transplanted to a city in northern Italy has the same ability as *H. physodes* and *P. furfuracea* to accumulate various metals [52].

According to Meli et al. (2018), lichens more easily accumulate air pollutants because they get most of their nutrients from the air; moreover, they have slow-growing properties and long life spans [53]. This capacity of lichens to accumulate various metals—especially heavy/trace metals—must be rigorously considered when exploring their use as edible or medicinal plants [38,54]. Heavy metal accumulation along the food chain represents a potential threat to human health [55], disturbing numerous biochemical processes [56]. Therefore, permissible limits of heavy/trace metals in edible and medicinal plants were established, aiming for their safe use [55]. The World Health Organization (WHO, 1996) and the Food and Agriculture Organization of the United Nations (FAO) indicated the permissible limits for heavy metals in edible plants as follows: 0.5 µg/g arsenic (As), 0.02 µg/g cadmium (Cd), 1.3 µg/g chromium (Cr), 0.01 µg/g cobalt (Co), 10 µg/g copper (Cu), 0.03 µg/g mercury (Hg), 0.1 µg/g nickel (Ni), 2 µg/g lead (Pb), 0.03 µg/g vanadium (V), and 50 µg/g zinc (Zn) [57,58].

Simkova and Polesny (2015) mentioned *U. barbata* as a culinary plant in the Balkan zone, consumed as mush and ingredients of bread [59]. Our metal content values, correlated to the previously mentioned limits, show that Cr (1.002 ± 0.008 µg/g), Cu (1.523 ± 0.013 µg/g), Pb (1.296 ± 0.007 µg/g), and Zn (20.536 ± 0.125 µg/g) are lower than corresponding permissible limit values. Other heavy metals registered higher contents than permissible ones: Hg (0.671 ± 0.020 µg/g), Ni (0.449 ± 0.011 µg/g), and V (0.241 ± 0.004 µg/g). The other toxic metals (As, Cd, and Co) were non-detected.

Dobrescu et al. (1993) displayed various *U. barbata* therapeutical properties, mentioning that it is used in traditional medicine as an antiseptic (in the USA and Spain) and for ameliorating the symptoms associated with various diseases: insomnia, bleeding, nausea, jaundice, and whooping cough (in Europe) [60]. The permissible limits for heavy/trace metals in medicinal plants are higher than in edible ones. According to the European Pharmacopoeia [55], the following permissible limits are available: 1 µg/g Cd, 2 µg/g Cr, 0.1 µg/g Hg, and 5 µg/g Pb. The Pharmacopoeia of the People’s Republic of China recommends 2 µg/g As, 0.3 µg/g Cd, 20 µg/g Cu, and 0.2 µg/g Hg [61]. According to the admissible limits of the WHO (2012) and the US Food and Drugs Administration (FDA), the following values were established: 10 µg/g As, 0.2 µg/g Cd, 20 µg/g Cu, 1 µg/g Hg, 10 µg/g Pb, and 50 µg/g Zn [61]. Our results showed that autochthonous *U. barbata* contained all heavy/trace metal contents in lower amounts than these permissible limits. However, the mercury content (0.671 ± 0.020 µg/g) was lower than the WHO’s and the FDA’s permissible limits (1 µg/g), but was over the ones mentioned in the European Pharmacopoeia (0.1 µg/g) and the Pharmacopoeia of the People’s Republic of China (0.2 µg/g).

The secondary metabolites display significant roles in lichens’ metal tolerance. These compounds are in vitro chelators of cations, including heavy metals. Bačkor and Fahselt (2004) [4] found that usnic acid may be associated with Cu, Ni, Fe, and Al in *Cladonia pleurota*. Another team of researchers found that usnic acid does not protect the cells of the photobiont *Trebouxia erici* against the toxic effect of Cu in a culture medium; both usnic acid and Cu became phytotoxic and inhibited photobiont growth, viability, and chlorophyll fluorescence [62]. Although metal complexes with secondary metabolites of lichens have been reported several times, their impacts on metabolic processes is far from wholly clarified [63].

Lichen secondary metabolites are also known for their antimicrobial activities. Antonenko et al. (2019) [64] described usnic acid as calcium ionophore [65] and protonophore, deciphering its antimicrobial mechanism. They highlighted the essential role of all OH groups of UA in protonophore potential [64], proving their abilities to induce membrane potential dissipation in isolated liver mitochondria and bacterial cells [65].

Polyphenols can generate aggregates in the cell wall of Gram-positive bacteria. They can also induce microscale grooves in the Gram-negative bacterial cell envelopes [66,67]. Moreover, in disturbing the folate metabolism, various polyphenols could inhibit ergosterol production in their action against *Candida* sp. [68]. In our previous study, using the HPLC method, we identified six phenolic acids (caffeic acid, p-coumaric acid, ellagic acid, chlorogenic acid, gallic acid, and cinnamic acid) in *U. barbata* ethanol extract. Chlorogenic acid, gallic acid, and p-coumaric acid were also found in *U. barbata* water extract [37].

As macromolecular polyphenols, tannins contain many phenolic hydroxylic groups, and this structural feature underlies their antimicrobial action [69] by various mechanisms. Tannins can interact with bacterial cell membranes to mediate antibacterial effects by affecting the membrane potential or increasing permeability [70]. They also can inhibit bacterial cell wall synthesis by directly binding to it or inactivating the enzymes. Furthermore, tannins seem to affect bacterial growth in several ways, such as inhibition of extracellular microbial enzymes, direct action on microbial metabolism through inhibition of oxidative phosphorylation, or deprivation of the substrates required for microbial growth [70]. For example, the o-dihydroxy phenyl groups in tannin molecules are involved in the chelation of ferric ions [71]; therefore, iron cannot be available to bacteria, leading to the inhibition of bacterial growth due to iron deprivation.

The antimicrobial effects of various secondary lichen metabolites could explain the antibacterial and antifungal activities of *Usnea barbata* extracts. Our data showed that *S. aureus* was susceptible to UA in a dose-dependent manner and resistant to all UBDE; on the other hand, *E. casseliflavus* was highly susceptible to UA and all UBDE. On *S. pyogenes,* only UBM had an antibacterial effect; this bacterium was resistant to UA and other UBDE. Usnic acid, UBEA, UBE, and UBA manifested antibacterial action against *S. pneumoniae;* only UBM was ineffective. On *P. aeruginosa*, all UBDE were significantly active, and UA proved to have an antibacterial effect in a dose-dependent manner. Contrariwise, the other Gram-negative bacteria, *E. coli* and *K. pneumoniae*, were highly resistant to UA and all UBDE. Finally, *C. albicans* was susceptible in a dose-dependent manner to UBM and UBE, and *C. parapsillosis* showed a significant susceptibility to UA and a high resistance to all UBDE.

These results can be correlated with *U. barbata*’s phytochemical and mineral profile, and explained based on the various metabolites with antibacterial and antifungal activity quantified in each UBDE. Thus, the solubility of usnic acid increases in order: water, ethanol, methanol, acetone, and ethyl acetate [72]; usnic acid and UBEA showed similar antibacterial activities in our study because this lichen extract has the highest usnic acid content [46]. On *Candida* sp., we can observe that, when only usnic acid had considerable inhibitory activity, UBDE in ethyl acetate showed an insignificant effect; also, when usnic acid showed a low inhibition, UBEA presented no activity. UBDE in methanol, acetone, and ethanol contain different secondary metabolites with antimicrobial activity (usnic acid polyphenols and tannins); thus, their approximately similar antibacterial and antifungal effects, with low differences, can be explained. The various mechanisms involved in both activities can also justify the differences between antibacterial and antifungal effects. Water extract had only low polyphenol and tannin contents; thus, the fact that UBW did not show any inhibitory action on bacterial or fungal colony growth could be justified.

These data suggest the involvement of all metabolites in the antimicrobial activities of *U. barbata* extracts. Moreover, these effects may be due to the quantified metals in the dried lichen (Zn [73], Cu [74], and Fe [75]), which could also be found in UBDE. These metals could have their own activities or generate synergisms, potentiating other constituents’ antibacterial and antifungal effects.

Numerous authors have evaluated lichen extracts’ antibacterial and antifungal activities on various pathogens resistant to current antimicrobial medications; their results were similar to those obtained in this study.

For example, Shrestha et al. (2014) studied the antibacterial activity of 34 North American lichen species against *S. aureus, P. aeruginosa,* and *E. coli* [76]. While all lichen species tested showed antibacterial action on *S. aureus* and *P. aeruginosa, E. coli* was susceptible to only three species out of the 34 studied; only two tested *Usnea* species had no antibacterial activity on *E. coli*. *Usnea hirta* and *Usnea strigosa*, tested in this study, showed high antibacterial activity only against *S. aureus* and *P. aeruginosa.*

In another study, Jha et al. (2017) analyzed *S. aureus, K. pneumoniae,* and *C. albicans* susceptibility to the inhibitory actions of 84 lichen species from Nepal. Their results proved that seventeen extracts showed activities against *S. aureus* and 45 extracts against *K. pneumoniae;* twelve extracts showed inhibitory activities against both bacterial species [77]. Only three specimen extracts were active against *C. albicans*. The three tested *Usnea* species (*Usnea pectinata, Usnea bailey,* and *Usnea coraline*) showed no antimicrobial activity in their study.

Kumar et al. (2017) extracted protolichesterinic acid from *Usnea albopunctata* and studied its antifungal effect against four *Candida* species, including *C. albicans* and *C. parapsilosis;* the diameters of inhibition zones were 21 mm and 22 mm, respectively [78]. Our results also showed that the antifungal activity of usnic acid was higher against *C. parapsilosis* (20 mm) than against *C. albicans* (10 mm).

Rankovic et al. (2009) studied the antibacterial action of five lichen species, testing three different extracts for each lichen species: acetone, methanol, and water [79]. All lichen extracts in water showed no antibacterial activity; both extracts in methanol and acetone registered a high level of inhibition on bacterial strains. Singha et al. (2011), using the previously mentioned solvents for lichen extraction, reported that methanol extracts had the most intense antibacterial effects [80].

Analyzing the antimicrobial activity of lichen extracts, most studies were performed by testing 1–3 extracts of the same species in different solvents [81]. Other researchers compared the actions of lichens extracts using 1–3 solvents with those of the most active metabolites [82] contained by the tested species [83]. However, studies that evaluate more than three extracts in different solvents of the same lichen species [84] are in a much smaller number [85]; our study examined five *Usnea barbata* (L.) F.H. Wigg dry extracts, and the differences between their antimicrobial activities were analyzed in relationship with the phytoconstituents extracted with each solvent.

## 4. Materials and Methods

### 4.1. Lichen Samples

*U. barbata* thalli were harvested one by one from the branches of conifers in the Călimani Mountains—the highest Romanian volcanic mountains—in the early spring of 2020. The freshly collected lichens were separated from the impurities; then, they were dried at 18–25 °C in a herbal room, harbored from sunlight. Preservation of the dried lichens for an extended period was performed under the same conditions. The Department of Pharmaceutical Botany of the Faculty of Pharmacy, Ovidius University of Constanta accomplished the *U. barbata* identification using standard methods [86].

### 4.2. Elemental Analysis

The dried *U. barbata* lichen was used for ICP-MS elemental analysis, according to the European Pharmacopoeia 10.0 [87]; 23 elements were analyzed: Ca, Fe, Mg, Mn, Zn, Al, Ag, Ba, Co, Cr, Cu, Li, Ni, Tl, V, Mo, Pd, Pt, Sb, As, Pb, Cd, and Hg, using the ICP ability to generate charged ions from the element species within the lichen sample [87]. Thus, they are guided [88] into a mass spectrometer and separated according to their mass-to-charge ratio (*m*/*z*) [87].

#### 4.2.1. Equipment

The quadrupole inductively coupled plasma mass spectrometer was a NexION™ 300S (PerkinElmer, Inc., Hopkinton, MA, USA), with a triple cone interface and a four-stage vacuum system. This ICP-MS system is equipped with a universal cell with two gas lines (helium, ammonia, and methane) that allows operation in collision mode (helium) and reaction mode (ammonia/methane). The ICP-MS is equipped with a recirculating chiller (PerkinElmer, Shelton) and a peristaltic pumping system with acid-resistant tubing—0.38 mm interior diameter (id) tubing for sample introduction and 1.3 mm id for drain exclusive.

The samples were digested in mineralization vessels using Rotor 16HF100 in a PROSOLV microwave digestion system (Anton Paar GmbH, Graz, Austria) using a pressure-activated venting concept. The Directed Multimode Cavity (DMC) enables highly efficient turbo heating with one magnetron in a compact system combined with a turbo cooling system for rapid cooling from 180 °C to 70 °C.

#### 4.2.2. Dried Lichen Mineralization

A sample of 0.250 g ± 0.05 g homogenized dried lichen was weighed on an analytical balance Quintix^®^ 224-1CEU (Sartorius Lab Instruments GmbH & Co. KG, Göttingen, Germany). Then, it was added to the digestion vessel with 4 mL 65% HNO_3_ and 1 mL 30% H_2_O_2_ (Merck, KGaA, Darmstadt, Germany). After 30 min pre-reaction, the dishes were placed in the microwave digestion system. The selected parameters of the digestion process are registered in Table 6.

The reagent control (blank) was obtained by adding 4 mL 65% HNO_3_ and 1 mL 30% H_2_O_2_ in a sample-free Teflon tube and mineralizing it with the dried lichen. After the digestion program, the samples were transferred to 25 mL volumetric flasks and brought to the mark with ultrapure deionized water. The ultrapure water was obtained with a Simplicity^®^ UV Water Purification System (Merck Millipore, Burlington, MA, USA), equipped with a dual-wavelength UV lamp ensuring photo-oxidation of organic compounds and flow rates >0.5 L/min.

#### 4.2.3. Standard Solutions

Two different concentrations of standard elemental stock solutions (PerkinElmer, Inc, Hopkinton, MA, USA) were used. The mercury stock solution had a concentration of 10 μg/mL, and all other elements’ stock solutions’ concentrations were 1000 μg/mL. The Multi-Element Standard for ICP-MS Instrument Calibration was requested for NexION Setup Solution (Be, Ce, Fe, In, Li, Mg, Pb, U) and for NexION KED Mode Setup Solution (Be, Ce, Fe, In, Li, Mg, Pb, U). The preparation of the standard solutions and calibration standard solutions is detailed in the Appendix A. For each element calibration curve, different calibration standard solutions E1–E5 (Appendix A) with several concentrations (µg/L) were obtained (Table 7).

#### 4.2.4. Working Conditions

The ICP-MS elemental analysis was performed by the kinetic energy discrimination (KED) method, measuring unit = counts per second (CPS). The peristaltic pumping system was washed with each sample (35 s), followed by a read delay (15 s), and the analytical phase. Afterwards, the peristaltic pump was washed with ultrapure deionized water (45 s). All processes involved an operation speed = 20–24 rotations/minute (rpm).

#### 4.2.5. Specificity

Specificity was verified by determining the recovery of each analyzed element in spiked sample solutions. Sample spiking was performed at each element’s estimated LOQ (µg/L) value. According to the United States Environmental Protection Agency (USEPA), available spike recovery (%) values are included in the range 70–150% [89]. Spike recovery (%) was calculated according to the following Equation (1):(1)$$\mathrm{Spike}\;\mathrm{recovery}\;(\%)=\frac{concentration\;of\;spiked\;solution-concentration\;of\;sample\;solution}{theoretical\;concentration\;of\;spiked\;solution}\times 100$$

#### 4.2.6. Spike Solutions

The four spike solutions were prepared using standard elemental stock solutions (Appendix A), as follows:

Spike Sol. I (As, Pb, Cd, and Hg) 1 mg/L: Into a 20 mL volumetric flask were added: 0.2 mL 65% HNO_3_, 0.02 mL solution As 1000 mg/L, 0.02 mL solution Pb 1000 mg/L, 0.02 mL solution Cd 1000 mg/L, and 2 mL solution Hg 10 mg/L; then, the obtained solution was brought to the mark with ultrapure water;

Spike Sol. II (Ca, Fe, Mg, Mn, and Zn) 10 mg/L: Into a 50 mL volumetric flask were added: 0.5 mL 65% HNO_3_, 0.5 mL solution Ca 1000 mg/L, 0.5 mL solution Fe 1000 mg/L, 0.5 mL solution Mg 1000 mg/L, 0.5 mL solution Mn 1000 mg/L, and 0.5 mL solution Zn 1000 mg/L; then, the flask content was brought to the mark with ultrapure water;

Spike Sol. III (Al) 10 mg/L: Into a 50 mL volumetric flask were added 0.5 mL 65% HNO3, 0.5 mL solution Al 1000 mg/L, and ultrapure water up to the mark;

Spike Sol. IV (Ag, Ba, Co, Cr, Cu, Li, Ni, Tl, V, Mo, Pd, Pt, and Sb) 1 mg/L: Into a 50 mL volumetric flask were added: 0.5 mL 65% HNO_3_, 0.05 mL solution Ag 1000 mg/L, 0.05 mL solution Ba 1000 mg/L, 0.05 mL solution Co 1000 mg/L, 0.05 mL solution Cr 1000 mg/L, 0.05 mL solution Cu 1000 mg/L, 0.05 mL solution Li 1000 mg/L, 0.05 mL solution Ni 1000 mg/L, 0.05 mL solution Tl 1000 mg/L, 0.05 mL solution V 1000 mg/L, 0.05 mL solution Mo 1000 mg/L, 0.05 mL solution Pd 1000 mg/L, 0.05 mL solution Pt 1000 mg/L, and 0.05 mL solution Sb 1000 mg/L; then, the obtained solution was brought to the mark with ultrapure water.

#### 4.2.7. Spiked Solutions

Spiked solutions at LOQ (µg/L) estimated level of each element were prepared as follows: a sample of 0.250 g ± 0.05 g homogenized dried lichen was weighed on a Quintix^®^ 433 analytical balance 224-1CEU (Sartorius Lab Instruments GmbH & Co. KG, Göttingen, Germany). Then, it was added to the digestion vessel with 4 mL 65% HNO_3_ and 1 mL 30% H_2_O_2_ (Merck, Germany). Supplementarily, in the Teflon vessel, 0.025 or 0.125 mL spike solutions were added according to the data summarized in Table 8. After the digestion program, the samples were transferred to 25 mL volumetric flasks and brought to the mark with ultrapure deionized water.

In the spike recovery (%) calculation, we prepared two dilutions of the sample solution—1:10 (d = 10) and 1:100 (d = 100)—using 2.5 mL and 0.25 mL, respectively, of the mineralized sample solution. These were added into two 25 mL volumetric flasks. Then, the corresponding spike solutions from Table 8 were added and brought to the mark with ultrapure deionized water. All of these data are presented in Appendix A.

#### 4.2.8. Calibration Curves

The calibration curves were performed using five calibration standard solutions (E1–E5, µg/L) for each element (Appendix A). Using linear equations, the correlation coefficients (R^2^) were determined, with the admissibility condition requesting an R^2^ value > 0.99.

#### 4.2.9. Detection Limits and Quantification Limits

The detection limit (LOD) and quantification limit (LOQ) values for each element were calculated using the standard deviation (SD) for ten determinations of the first calibration standard solution E1 and the slope of the calibration curve (Appendix A), applying the following Equation (2):
(2)LOD = 3.3 × SD/Slope

The element content (Table 1), expressed as µg/g or mg/kg, was calculated according to the data from the calibration curve: *element concentration* (µg/L), the weighted sample amount—*m* (g), and the final volume of the sample solution—*V_f_* (mL), according to Equation (3):(3)$$\mathrm{Element}\;\mathrm{content}\;(µg/g)=\frac{\;element\;concentration\;\left(\frac{µg}{L}\right)\times V_{f}\left(\mathrm{mL}\right)}{m\left(g\right)*1000}$$

#### 4.2.10. Data Analysis, Software

Data obtained were processed with Syngistix software (PerkinElmer, Inc., Hopkinton, MA, USA) Version 2.3. for ICP-MS. Elemental analysis was performed in triplicate, and the results are expressed as mean (n = 3) ± SD.

### 4.3. U. barbata Dry Extracts—Preparation

In a Soxhlet continuous reflux system, the dried lichen was ground into a powder; 20 g of dried lichen was extracted for 8 h with 150 mL solvent: ethyl acetate, methanol (Chemical Company, Iasi, Romania), acetone, ethanol (Chimreactiv S.R.L., Bucharest, Romania), and water [46]. The extraction temperature was different for each extract, being around the boiling point of each solvent (Table 1). After filtration with filter paper, UBW was concentrated on a Rotavapor R-215 with a vacuum controller V-850 (BÜCHI Labortechnik AG, Flawil, Switzerland), and lyophilized with a freeze-dryer Christ Alpha 1-2L (Martin Christ Gefriertrocknungsanlagen GmbH, Osterode am Harz, Germany) connected to a vacuum pump RZ 2.5 (VACUUBRAND GmbH, Wertheim, Germany) [46]. For UBEA, UBA, UBE, and UBM, the rotary evaporator TurboVap 500 (Caliper Life Sciences Inc, Hopkinton, MA, USA) evaporated the correspondent solvents [41]. Next, each extract was kept for 16 h in a chemical exhaust hood for optimal solvent evaporation. All obtained *U. barbata* dry extracts were transferred to sealed-glass containers and stored in the freezer (Sirge^®^ Elettrodomestici—S.A.C. Rappresentanze, Torino, Avigliana (TO) Italy) at −24 °C until processing [46].

### 4.4. Determination of the Usnic Acid Content

Usnic acid was determined in *U. barbata* dry extracts by ultra-high performance liquid chromatography (UHPLC) [46]. All UBDE were re-dissolved in acetone, ethyl acetate, ethanol, methanol, and water, and diluted to 1:50 with DMSO [46]. The PerkinElmer^®^ Flexar^®^ FX-15 UHPLC system fitted with a Flexar FX PDA-Plus photodiode array detector was the platform for this analysis. The Brownlee Analytical C18 column, with an inner diameter of 4.6 mm and a length of 150 mm, was filled with 5 µm superficially porous particles. Working conditions consisted of: flow = 1.5 mL/min; temperature in the column compartment = 25 °C; injection volume = 20 µL; and analysis time: 10 min. The mobile phase was an isocratic system of methanol/water/glacial acetic acid (80:15:5). After elution from the column, the separated compounds were analyzed at a wavelength of 282 nm [46].

### 4.5. Determination of the Total Polyphenol Content

The total polyphenol content was determined with Folin-Ciocâlteu reagent (phosphomolybdotungstic acid) using pyrogallol as the standard [46]. The method was described in our previous study [46] and TPC values were expressed as mg of pyrogallol equivalents (PyE) per g UBDE. For this analysis, 5 mL of each UBDE (A1–A5) filtered through 99 filter paper were added to five volumetric flasks of 25 mL and completed up to the sign with the same solvent. Then, 2 mL of each previously obtained solution (B1–B5) were added into five volumetric flasks of 25 mL, with 1 mL of Folin-Ciocâlteu reagent, 10 mL water, and 290 g/L of Na_2_CO_3_ solution, up to the mark. After 30 min of reaction at room temperature, the absorbance values (each value was noted with A1 in the calculation formula) were determined at 760 nm, using a Jasco V630 UV-Vis Spectrophotometer (JASCO Corporation, Tokyo, Japan) with Spectra Manager™ software [46].

### 4.6. Determination of the Tannin Content

As per our previous study [46], the tannin content was determined using a three-phase procedure: determination of TPC in different UBDE extracts by the Folin-Ciocâlteu method, absorption of tannins on standardized hide-powder, and determination of the phenolic compounds in the solution remaining after the second phase. The quantification of the molybdenum oxide’s blue coloration intensity was determined by spectrophotometry at 760 nm, and the difference between both determinations even represents the tannin content [46].

### 4.7. Antimicrobial Activity

The antibacterial and antifungal activities were evaluated by an adapted disc diffusion method from the Clinical and Laboratory Standard Institute (CLSI) [90,91].

#### 4.7.1. Microorganisms and Media

All microorganism strains were obtained from the American Type Culture Collection (ATCC) for our study. They were identified at the Department of Microbiology and Immunology, Faculty of Dental Medicine, Ovidius University of Constanta. The Gram-positive bacteria were *Staphylococcus aureus* (ATCC 29213), *Enterococcus casseliflavus* (ATCC 700327), *Streptococcus pyogenes* (ATCC 19615), *Streptococcus pneumoniae* (ATCC 49619), and the group of Gram-negative bacteria included *Escherichia coli* (ATCC 25922), *Klebsiella pneumoniae* (ATCC 13883), and *Pseudomonas aeruginosa* (ATCC 27853). The antifungal activity evaluation was performed using *Candida albicans* (ATCC 14053) and *Candida parapsilosis* (ATCC 22019).

Mueller-Hinton agar with 5% defibrinated sheep blood (Thermo Fisher Scientific, GmbH, Dreieich, Germany) was used as a culture medium for both *Streptococcus* sp. [90]. The other bacterial strains were maintained in Mueller Hinton agar (Thermo Fisher, Dreieich, Germany). For both *Candida* sp., Sabouraud 4% Glucose Agar (Merk KGaA, Darmstadt, Germany) was selected as the culture medium.

#### 4.7.2. Inoculum Preparation

The bacteria inoculum was prepared by the direct colony suspension method (CLSI) [90]. Thus, a 0.9% saline suspension of bacterial colonies selected from a 24 h agar plate was performed, according to the 0.5 McFarland standard, with around 10^8^ CFU/mL (CFU = colony-forming unit) [90]. The yeast inoculum was prepared using the same method, adjusting the saline suspension of fungal colonies to the 0.5 McFarland standard, with 10^6^ CFU/mL [92].

#### 4.7.3. Disc Diffusion Method

*Usnea barbata* dry extracts and usnic acid were dissolved in 0.1% DMSO and applied on Whatman^®^ filter paper discs (6 mm, Merk KGaA, Darmstadt, Germany). The solvent (0.1% DMSO) was the negative control and UA in 0.1% DMSO (129 mg/mL) was the positive control for UBDE [46]. The weighted mass values for UA and each UBDE were similar to those used in our previous study to evaluate their cytotoxic activity by brine shrimp lethality assay [46]. Therefore, the concentrations of the sample UBDE solutions were as follows: 172 mg/mL UBEA, 162 mg/mL UBA, 161 mg/mL UBE and UBM, and 160 mg/mL UBW [46]. Each filter paper disc was impregnated with 10 µL solution. For antimicrobial activity evaluation, blank antibiotic discs (6 mm)—Levofloxacin 5 µg, Tetracycline 30 µg, and antifungal ones—Voriconazole 1 µg and Fluconazole 15 µg (Oxoid, Thermo Fisher Scientific GmbH, Dreieich, Germany) were used. The blank discs were maintained in a freezer at −14 °C and incubated at room temperature for 2 h before analysis.

Each inoculum was applied with a sterile cotton swab over the entire surface of the plate with the suitable culture media. After 15 min of drying, the filter paper discs were applied to the inoculated plates. The plates were incubated for 24 h at 37 °C.

#### 4.7.4. Reading Plates

Examining the plates after 24 h, circular zones of a microorganism growing inhibition around several discs could be observed. The results of the disc diffusion assay are expressed in the inhibition zone size (IZs) diameter, measured in mm. IZs values [90] quantify the level of susceptibility of microbial strains after 24 h incubation.

#### 4.7.5. Interpretation of DDM Results

Usnic acid and UBDE IZs were compared to the IZs values of the positive controls represented by the blank antibiotic/antifungal discs. In DDM, IZs values inversely correlate with minimum inhibitory concentrations (MIC) from standard dilution tests [90]. According to CLSI, the interpretive categories are as follows: Susceptible (“S”), Intermediate = dose-dependent susceptibility (“I”), and Resistant (“R”) [90].

#### 4.7.6. Data Analysis, Software

The analyses were performed in triplicate, and the results are expressed as mean (n = 3) ± SD, calculated by Microsoft 365 Office Excel. The *p*-values were determined with the one-way ANOVA test. The differences between the mean values were considered significant when the *p*-value was <0.05.

## 5. Conclusions

Our study’s novelty consists of ICP-MS analysis of 23 metals of *Usnea barbata* (L.) F.H. Wigg. from an unpolluted zone in the Călimani Mountains, Romania. The obtained data were analyzed in correlation with the specific properties of this lichen habitat zone and the trace/heavy metal content in the soil. Moreover, we compared the heavy metal content in *U. barbata* with permissible limits of toxic elements and found that only mercury was over the limit mentioned in the European Pharmacopoeia for medicinal plants.

Another original aspect of our research involves evaluation and comparative analysis of the antimicrobial actions of usnic acid and five dry extracts of *U. barbata* in different solvents against various pathogens known for their resistance to common antibacterial and antifungal drugs.

The obtained results could enrich the existing information in the scientific databases—which must be constantly updated—by the metal content and antimicrobial potential of autochthonous lichen from a peat bog zone of the highest Romanian volcanic mountains.

The antimicrobial study highlights that usnic acid and various *U. barbata* dry extracts (except water extract) have proven significant antibacterial and antifungal activities. The most susceptible microorganisms were Gram-positive bacteria. The antimicrobial potential of *U. barbata* dry extracts also demonstrates a moderate correlation with metabolite content.

Our results suggest that further research can be aimed at advanced studies on the antibacterial and antifungal potential. Future studies could decipher the biochemical mechanisms and establish suitable doses of *U. barbata* extracts for an optimal antimicrobial effect.

## Figures and Tables

**Figure 1 plants-11-00032-f001:**
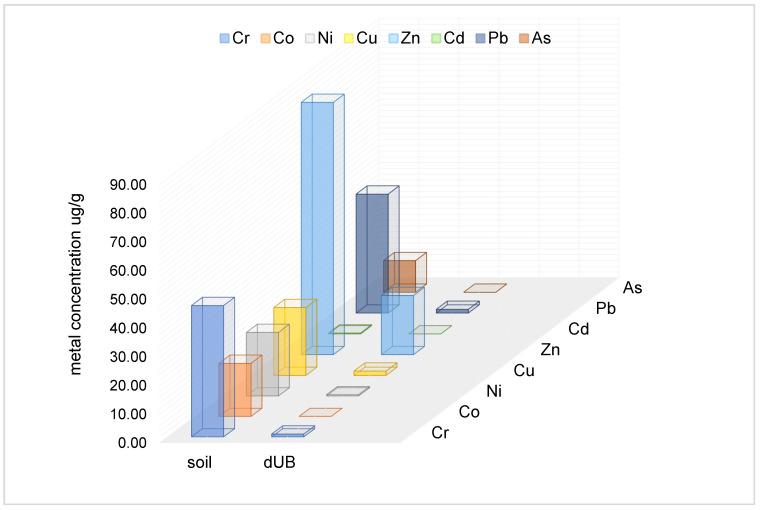
Correspondence between trace/heavy metal contents in soil and native dried *Usnea barbata* (L.) Weber ex F.H. Wigg (dUB).

**Table 1 plants-11-00032-t001:** Mineral composition of *Usnea barbata* (L.) Weber ex F.H. Wigg dried lichen.

Element	Content (µg/g)	LOQ (µg/g)
Ag	ND	0.100
Al	87.879 ± 1.152	1.000
As	ND	0.100
Ba	3.782 ± 0.052	0.100
Ca	979.766 ± 12.285	5.000
Cd	ND	0.100
Co	ND	0.100
Cr	1.002 ± 0.008	0.100
Cu	1.523 ± 0.013	0.100
Fe	52.561 ± 2.582	5.000
Li	ND	0.100
Mg	172.721 ± 0.647	5.000
Mn	101.425 ± 1.423	5.000
Ni	0.449 ± 0.011	0.100
Pb	1.296 ± 0.007	0.100
Tl	ND	0.100
V	0.241 ± 0.004	0.100
Zn	20.536 ± 0.125	5.000
Hg	0.671 ± 0.020	0.100
Mo	ND	0.100
Pd	ND	0.100
Pt	ND	0.100
Sb	ND	0.100

The analysis was performed in triplicate. Results are presented as mean ± standard deviation (SD). LOQ—quantification limit (µg/g); ND—non-detected, Ag—silver, Al—aluminum, As—arsenic, Ba—barium, Ca—calcium, Cd—cadmium, Co—cobalt, Cr—chromium, Cu—copper, Fe—iron, Li—lithium, Mg—magnesium, Mn—manganese, Ni—nickel, Pb—lead, Tl—thallium, Zn—zinc, Hg—mercury, Mo—molybdenum, Pd—palladium, Pt—platinum, Sb—antimony.

**Table 2 plants-11-00032-t002:** Extraction conditions and secondary metabolite content of various dry extracts of *Usnea barbata* (L.) Weber ex F.H. Wigg.

UBDE	Solvent	Temperature of Extraction (°C)	Yield %	UAC (mg/g UBDE)	TPC (mgPyE/g UBDE)	TC (mg PyE/g UBDE)
UBEA	Ethyl acetate	75–80	6.27	376.73	42.40	24.4
UBA	Acetone	55–60	6.36	282.78	101.09	3.85
UBE	Ethanol	75–80	12.52	127.21	67.30	14.70
UBM	Methanol	65	11.29	137.60	70.70	9.99
UBW	Water	95–100	1.98	0.00	45.80	1.31

UBDE—*U. barbata* dry extracts, UBEA—*U. barbata* dry extract in ethyl acetate, UBA—*U. barbata* dry extract in acetone, UBE—*U. barbata* dry extract in ethanol, UBM—*U. barbata* dry extract in methanol, UBW—*U. barbata* dry extract in water, UAC—usnic acid content, TPC—total polyphenol content, TC—tannin content, mg PyE/g UBDE—mg equivalents of pyrogallol per g UBDE.

**Table 3 plants-11-00032-t003:** Antibacterial activity of *Usnea barbata* (L.) Weber ex F.H. Wigg dry extracts on Gram-positive and Gram-negative bacteria.

Sample	UA	UBEA	UBA	UBE	UBM	UBW	LEV		TET	
Diffusible amount (µg)	1290 Spot area of growing inhibition (mm)	1720	1620	1610	1610	1600	5		30	
Bacteria	IZs (mm)									
*Staphylococcus aureus*	16.00 ± 0.82	14.33 ± 0.94	12.66 ± 1.24	11.66 ± 0.94	13.00 ± 0.82	0	28.33 ± 2.49		25.66 ± 2.49	
	a *, k *	a *, f, m *	a *, f, n *	a *, f, o *	a *, f, r *		S	R	S	R
							≥19	≤15	≥19	≤14
							I = 18–16		I = 18–15	
*Enterococcus casseliflavus*	19.67 ± 1.70	20.33 ± 1.89	20.00 ± 2.94	20.00 ± 3.26	22.00 ± 0.82	0	25.00 ± 0.82		26.00 ± 1.63	
	b, k *	b, g, m *	b, g, n *	b, g, o *	b, g, r *		S	R	S	R
							≥17	≤13	≥19	≤14
							I = 16–14		I = 18–15	
*Streptococcus pyogenes*	12.00 ± 0.82	12.67 ± 1.25	10.00 ± 0.82	12.00 ± 1.63	20.00 ± 1.63	0	21.00 ± 1.63		27.00 ± 1.63	
	c *, k *	c *, h *, m *	c *, h *, n *	c *, h *, o *	c *, h *, r *		S	R	S	R
							≥17	≤13	≥23	≤18
							I = 16–14		I = 22–19	
*Streptococcus pneumoniae*	17.00 ± 1.63	17.00 ± 0.82	18.00 ± 0.82	18.00 ± 1.63	13.33 ± 0.94	0	22.00 ± 1.63		30,67 ± 2.05	
	d *, k *	d *, i *, m *	d *, i *, n *	d *, i *, o *	d *, i *, r *		S	R	S	R
							≥17	≤13	≥24	≤20
							I = 16–14		I = 23–21	
*Escherichia coli*	7.00 ± 0.82	0	0	0	0	0	31.00 ± 1.63		21.00 ± 0.82	
	k *	m *	n *	o *	r *		S	R	S	R
							≥17	≤13	≥15	≤11
							I = 16–14		I = 14–12	
*Klebsiella pneumoniae*	0	0	0	0	0	0	27.00 ± 1.63		20.00 ± 1.63	
	k *	m *	n *	o *	r *					
							S	R	S	R
							≥17	≤13	≥15	≤11
							I = 16–14		I = 14–12	
*Pseudomonas aeruginosa*	16.00 ± 0.82	17.33 ± 2.05	17.00 ± 0.82	20.00 ± 1.63	19.67 ± 1.25	0	21.00 ± 0.82		24.00 ± 1.63	
	e *, k *	e *, j, m *	e *, j, n *	e *, j, o *	e *, j, r *		S	R	S	R
							≥17	≤13	≥19	≤14
							I = 16–14		I = 18–15	

The analyses were performed in triplicate. The results are presented as mean (n = 3) ± standard deviation (SD). Levofloxacin and Tetracycline, with antibacterial effects against all bacterial species, were used for the interpretation of obtained results; their breakpoints (mm) were indicated: S—susceptibility zone, R—resistance zone, and I—intermediate, dose-dependent zone. IZs—the size of inhibition zone diameter (mm), UA—usnic acid, UBEA—*U. barbata* dry extract in ethyl acetate, UBA—*U. barbata* dry extract in acetone, UBE—*U. barbata* dry extract in ethanol, UBM—*U. barbata* dry extract in methanol, UBW—*U. barbata* dry extract in water, LEV—Levofloxacin, TET—Tetracycline. Different lower-case letters (a, b, c, d, e, f, g, h, i, j, k, m, n, o, and r) placed under IZs values show the series of IZs values compared for determination of p-value; the symbol * indicates statistically significant differences (*p* < 0.05).

**Table 4 plants-11-00032-t004:** Antifungal activity of *Usnea barbata* (L.) Weber ex F.H. Wigg dry extracts on *Candida* species.

Sample	UA	UBEA	UBA	UBE	UBM	UBW	FLUCZ		VORI	
Diffusible amount (µg)	1290	1720	1620	1610	1610	1600	25		1	
Fungi	IZs (mm)									
*Candida albicans*	10.00 ± 0.82	0	13.00 ± 1.63	15.33 ± 1.24	16.33 ± 2.05	0	32.33 ± 1.70		34.33 ± 1.25	
	a *, e *	a *, c *, f *	a *, c *, g *	a *, c *, h *	a *, c *, i *		S	R	S	R
							≥19	≤14	≥17	≤13
							I = 18–15		I = 16–14	
*Candida parapsilosis*	20.00 ± 1.63	7.00 ± 0.82	0	0	0	0	25.67 ± 2.49		30.67 ± 3.30	
	b *, e *	b *, d *, f *	b *, d *, g *	b *, d *, h *	b *, d *, i *		S	R	S	R
							≥19	≤14	≥17	≤13
							I = 18–15		I = 16–14	

The analyses were performed in triplicate. The results are presented as mean (n = 3) ± standard deviation (SD). With well-established breakpoints for the tested fungal strains, Fluconazole and Voriconazole were used to interpret the data obtained; I = Intermediate susceptibility zone; S and R = Susceptibility and Resistance breakpoints. IZs—the size of inhibition zone diameter (mm), UA—usnic acid, UBEA—*U. barbata* dry extract in ethyl acetate, UBA—*U. barbata* dry extract in acetone, UBE—*U. barbata* dry extract in ethanol, UBM—*U. barbata* dry extract in methanol, UBW—*U. barbata* dry extract in water, FLUCZ—Fluconazole, VORI—Voriconazole. Different lower-case letters (a, b, c, d, e, f, g, h, and i) indicate the IZs values compared for p-value determination; the symbol * shows statistically significant differences (*p* < 0.05).

**Table 5 plants-11-00032-t005:** Correlation between the antimicrobial activity of various *Usnea barbata* (L.) Weber ex F.H. Wigg dry extracts and the metabolite content, displaying linear equations and correlation coefficient values.

Bacteria	UAC	TPC	TC
*Staphylococcus aureus*	y = 0.0314x + 4.53	-	-
	R^2^ = 0.6187	-	-
*Streptococcus pneumoniae*	y = 0.039x + 6.0622	-	-
	R^2^ = 0.5571	-	-
*Candida albicans*	-	y = 0.2601x − 8.0934	-
	-	R^2^ = 0.5523	-
*Candida parapsilosis*	y = 0.0156x − 1.4826	-	y = 0.2796x − 1.6342
	R^2^ = 0.5342	-	R^2^ = 0.6766

UAC—usnic acid content, TPC—total polyphenol content, TC—tannin content, R^2^—correlation coefficient.

**Table 6 plants-11-00032-t006:** Dried lichen digestion conditions.

Step	Temperature (°C)	Power of Microwave Digestion System (W)	Time (min)	Fan Level
Power ramp	-	1450	15	1
Power hold	180	1450	45	1
Cooling	70	0	-	3

**Table 7 plants-11-00032-t007:** Concentrations (µg/L) of calibration standard solutions E1–E5 for different elements.

Element	E1 (µg/L)	E2 (µg/L)	E3 (µg/L)	E4 (µg/L)	E5 (µg/L)
As, Pb, Cd, Hg	1	5	10	15	25
Ca, Fe, Mg, Mn, Zn	50	100	200	300	500
Al	10	50	100	150	200
Ag, Ba, Co, Cr, Cu, Li, Ni, Tl, V, Mo, Pd, P, Sb	1	5	10	50	100

**Table 8 plants-11-00032-t008:** Preparation of spiked solutions.

Element (Spike)	Spike Solution		Spiked Solution Theoretical Concentration (µg/L)
	Concentration (mg/L)	Volume (mL)	
As, Pb, Cd, Hg	1	0.025	1
Ca, Fe, Mg, Mn, Zn	10	0.125	50
Al	10	0.025	10
Ag, Ba, Co, Cr, Cu, Li, Ni, Tl, V, Mo, Pd, Pt, Sb	1	0.025	1

Ag—silver, Al—aluminum, As—arsenic, Ba—barium, Ca—calcium, Cd—cadmium, Co—cobalt, Cr—chromium, Cu—copper, Fe—iron, Li—lithium, Mg—magnesium, Mn—manganese, Ni—nickel, Pb—lead, Tl—thallium, Zn—zinc, Hg—mercury, Mo—molybdenum, Pd—palladium, Pt—platinum, Sb—antimony.

## Data Availability

Data are contained within the article and Appendix A.

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
