# Peer review of "Elemental Analysis and In Vitro Evaluation of Antibacterial and Antifungal Activities of Usnea barbata (L.) Weber ex F.H. Wigg from Călimani Mountains, Romania"

_plants, 2021, doi:10.3390/plants11010032_

Round 1

Reviewer 1 Report

The main goal of the study was the examination of five Usnea barbata (L.) F.H.Wigg dry
extracts. The differences in antimicrobial activities stemmed from different secondary
metabolite compositions due to the application of five different extracting agents (acetone,
ethyl acetate, methanol, ethanol, water). The metabolite content was determined afterward and
antimicrobial activity was determined using DDM. Since lichen can efficiently absorb macro
and microelements the metals were determined along various UBDEs. The study, therefore,
employs the ICP-MS analysis of 23 metals in Usnea barbata (L.) F.H. 536 Wigg. from an
unpolluted zone in Calimani Mountains, Romania. A correlation analysis was performed
between the specific properties of this lichen habitat zone and the trace/heavy metals content
in the soil. The study determined the antimicrobial effect of usnic acid and five extracts. With
exception of the water extract, the rest seemed to exhibit significant antimicrobial potential.
The experimental part of the study was well designed and had proper analytical tools and data
analysis. The introduction and references did not have major flaws. Some revisions will be
needed in the Discussion and major revisions in the Results including captions in figures and
tables. I had trouble understanding many sentences due to confusing syntax and choice of
words. I would recommend thorough editing. The names of microorganisms were not always
written in italics and there were inconsistencies in writing units for °C (0C or 0 C). The identity
of compounds seems to remain unknown. What are the obstacles in the determination of the
identity of novel compounds in the lichen extracts and is it feasible to link the antimicrobial
activity to novel compounds?
Editing (suggestions)
13-15 rows not aligned
39 replace were non-detectable/were bellow detection limits
43 reolace meaningfully resistant/insensitive/not susceptible
52 add comma disease, and ameliorate
117 correct the microbes' affinity/the microbial affinity
155-156 correct element registered in Table 1/element presented in Table 1
172-173 correct the element contents (μg/g) were registered in relationship with R%
and LOQ (μg/g)/ the element contents (μg/g), R% and LOQ (μg/g) were reported
174-175 delete commas followed, in decreasing order, by Mg/ followed in decreasing
order by Mg
178 delete and, add comma levels, and Hg (0.671 ± 0.020 μg/g) and V(0.241
±/levels, Hg (0.671 ± 0.020 μg/g), and V(0.241 ±
179 correct reported insignificant amount/were present in insignificant amount
180 correct registered in Table 2 reveal those/shown in Table 2 demonstrated that
those
184 correct were synthesized in Table 3/were summarized in Table 3
187 correct Extraction parameters/ Extraction conditions
187 delete s metabolites/metabolite
194 correct strains growing/strain growth
194 delete registered
195 correct registered/observed
195-196 delete in all the plates with bacteria cultures around
197-198 the sentence has to be rewritten
200 correct slowly increase/increased
209 ? the meaning is not clear Reported to LEV,...
211 unclear meaning registered
214 delete slowly
216 replace reported/displayed
220-221 the sentence has to rewritten
223 „instead“ is not used correctly
227 the meaning is not cleare Tetracycline (TET) were used to interpret the DDM
results.
Table 4 correct bacterial cell line/bacteria
231 incorrect use reported
Fig1 the legend needs better choice of words
Fig1b has larger font on x-axis
234 word choice registered
236 word choice involving
237-239 the sentence has to be rewritten
244-256 word choice considered, highlights, reported, induced, registered
258-261 table description is not clear
Table 5 replace fungal strain line/fungi
264-265 syntax/sentence is not clear
267-268 syntax/sentence is not clear
284 sentence is not clear precipitation range belongs a year in this zone is 600-800 mL
289 delete slowly
293 correct represented/presented
295 correct mention/mentioned
299 delete s others/other
302 add article in the United States
323 word choice revised
334 word choice differently
335-337 sentence structure and meaning is not clear
339-341 syntax
349-350 syntax explained concerning the metabolites content
357-358 understanding difficulty, the sentence is wordy and not well structured
370-376 microbial strains have to written in italics
417 wrog word choice oriented
515 correct accomodated/incubated
526 word choice were interpreted concerning
558 word choice 3 replications/replicates

Reviewer 2 Report

The article is suitable for plants. Please find the minor comments inside the article. 

Reviewer 3 Report

Dear authors, 

Please see the comments below -->

line 45 - change the 'slowly higher' I think you meant 'slightly'

in the keywords - use the full latin name 

line 56 - paragraph not separrated

line 127- remove 'several'

Table 1- I dont think that the calibration data is needed in the article, this can be added to the supplement containing other calibration related data. In general such data is shown in articles where methods are developed and validated

In results section check that the latin names ar in italics several of the bacterial names are not.

You have measured the contents of usnic acid and total polyphenolics, however, the methods used for the charaterisation have not been described - this must be added.

the graphs in figure 1 are rather difficult to understand at first glance. There are a lot of abbreviations etc in the graphs - I would suggest making them easier for the reader to engage, since figures are the first thing that is beeing loked at when skipping through the article.

The TPC content was evaluated also in the acetone and ethyl acetate extracts, but those are not common solvents to be used for the extraction of polyphenolics, they are more often used for the extraction of condensed polyphenolics, for example tannins, procyanidins. 

Also, looking at Figure 1 it seems that the compounds responsible for the anti-microbial activity are neither the usnic acid nor the polyphenolics. in the lines 240-242 you are saying that there is correlation, but it seems that the graphs are showing that the lower concentration of TPC and UAC has higher antimicrobial activity.

The tables are also rather difficult to read... To the authors they might seem logical and easy to understand, but you have to think about the presentation and the reader when preparing a manuscript, so that difficult data could be easily understood without the help of the text and looking up abbreaviations.

Make the reference list so that the first line is not with indent and the following lines are, it looks messy this way.

Latin names must be italic also in the reference list, go through all of them...

Some references are missing a dot after the journal name

Overall the article is good, several things have to be adressed. I did not notice any mistakes in the use of English, but it is always beneficial to go through the manuscript once more. The first thing I noticed was that it is hard to justify why have you done the metal analysis and then antimicrobial analysis, since only a few metals (ones you have not measured) have anti-microbial effects. In the case where you are looking at antimicrobial activity it would be more benefficial to look at the seccondary metabolite profile, depending on the extract polarity (either use LC or GC to analyse the possible bioactive compounds).  Ideally you could put some work into the metal part of the article and publish it as a separate article and then do the same with the seccondary metabolites and antimicrobial activity. 

Round 2

Reviewer 1 Report

Some minor revisions are required. Suggestions:

45-46   The inhibitory activity of UBDE and UA was slightly higher in Candida albicans than Candida parapsilosis.

46        correct             Parapsilosis/parapsilosis

101      replace             For the last/For the latter (?)

111-112           replace             Many researchers proved that the extracts of Usnea lichens in different solvents had inhibitory activity on pathogens known for the antimicrobial resistance.

126      correct             another previous study using six Usnea ....

291      correct             other trace and heavy metals

340      correct             symptoms

346      replace             registered/established

352      add comma      In lichen, metal

397      add "s"                increases

414-435                       the name of microorganisms should be abbreviated such as C. albicans...

434      correct             ...also showed that antifungal activity of usnic acid was higher against ...

446      correct             extracts using different solvents

449      correct             extracted with each

455      correct             25 °C

Author Response

Please, see the attachments.

Reviewer 3 Report

Thank you for considering the improvements to the article and implementing them into the manuscript.

Good luck!
